# Effects of Different Yeast Selenium Levels on Rumen Fermentation Parameters, Digestive Enzyme Activity and Gastrointestinal Microflora of Sika Deer during Antler Growth

**DOI:** 10.3390/microorganisms11061444

**Published:** 2023-05-30

**Authors:** Weili Sun, Hongpeng Shi, Chengyan Gong, Keyuan Liu, Guangyu Li

**Affiliations:** 1College of Animal Science and Technology, Qingdao Agricultural University, Qingdao 266109, China; tcsswl@163.com (W.S.);; 2Institute of Special Animal and Plant Sciences of CAAS, Changchun 130112, China; 3University of Chinese Academy of Sciences, Beijing 100049, China

**Keywords:** sika deer, yeast selenium, fiber digestion, *Prevotella ruminicola*, *Fibrobacter succinogenes*

## Abstract

The aim of this experiment was to study the effects of different selenium supplemental levels on rumen fermentation microflora of sika deer at the velvet antler growth stage. A total of 20 5-year-old, healthy sika deer at the velvet antler growth stage with an average body weight of (98.08 ± 4.93) kg were randomly divided into 4 groups, and each group was fed in a single house. The SY1 group was the control group, and the SY2 group, SY3 group and SY4 group were fed a basal diet supplemented with 0.3, 1.2 and 4.8 mg/kg selenium, respectively. The pretest lasted for 7 days, and the formal trial period lasted for 110 days. The results show that: At the velvet antler growth stage, the digestibility of neutral detergent fiber and acid detergent fiber of sika deer in the SY2 group was significantly higher than that in the control group (*p* < 0.01). The digestibility of cellulose and crude fiber of sika deer in the SY2 group was significantly higher than those in the SY3 and SY4 groups (*p* < 0.01) and significantly higher than that in the control group (*p* < 0.05). The contents of acetic acid and propionic acid in the rumen fluid of sika deer in the SY2 group were significantly higher than those in the SY1 group (*p* < 0.05). Digestive enzyme analysis of rumen fluid at the velvet antler growth stage showed that the activity of protease in rumen fluid in the SY2 group was significantly lower than those in the SY1 group and SY4 group (*p* < 0.05). The relative abundance of *Fibrobacter succinogenes* in the SY2 group was significantly higher than that in the SY1 group (*p* < 0.05) and extremely significantly higher than those in the SY3 and SY4 groups (*p* < 0.01). Correlation analysis between yeast selenium level and bacterial abundance showed that the yeast selenium content in rumen fluid was significantly positively correlated with *Butyrivibrio* and *Succiniclasticum* (*p* < 0.01). Further verification of bacterial flora functioning showed that the SY2 group was more inclined to the degradation and utilization of fiber. In conclusion, 0.3 mg/kg selenium supplementation can increase the abundance of *Prevotella ruminicola* and *Fibrobacter succinogenes* in the rumen of sika deer and improve the degradation of fibrous substances by mediating the catabolite repression process.

## 1. Introduction

Velvet antler from sika deer, as a precious traditional Chinese medicine, is used in fields such as medicine, health care and skincare. Selenium is an essential trace element for animals, and selenium deficiency can easily lead to bone diseases. As cartilage tissue, deer antler is also susceptible to the influence of selenium. Studies have shown that selenium can affect the structure of rumen microflora and improve the digestibility of dietary fiber [1]. Dietary yeast selenium can increase the contents of propionic acid and total volatile fatty acids in rumen fluid [2], which is conducive to promoting rumen fermentation of ruminants [3]. Dietary selenium supplementation can promote rumen microflora colonization [4] and improve rumen microbial flora activity and digestive enzyme activity [5]. Selenium assimilation and reduction reactions occur under the action of rumen microorganisms, and selenium enters amino acids by replacing sulfur atoms and participates in the selenoprotein synthesis reaction of microflora [4].

Sika deer mainly digest fiber in the rumen. In addition, the environment in the rumen determines the utilization rate of dietary fiber [6]. As an essential trace element in animals, selenium can promote nutrient absorption, reduce oxidative stress and regulate the level of gastrointestinal flora in vivo [1]. Domestic and foreign scholars have studied the addition of organic selenium and inorganic selenium to diets, respectively. It has been found that organic selenium has the advantages of less toxicity and being easier to absorb by animals compared with inorganic selenium (Ferreira RLU, 2021). The results showed that dietary selenium can regulate the rumen environment, decompose fiber which is not easy to digest into easily absorbed nutrients, significantly improve fiber digestibility and volatile fatty acid production and improve the economic benefits of diets.

Yeast selenium, also called selenium-enriched yeast, is an organic selenium source developed using yeast. It is produced by enriching selenium in the cell protein structure of growing yeast. Yeast selenium is a kind of organic selenium prepared by yeast through biotransformation during fermentation, which is essentially different from some selenium preparations prepared by chelating inorganic selenium with small peptides or amino acids [7]. The production principle of yeast selenium is to select yeast strains with specific high-sulfur requirements and provide them with a culture medium rich in inorganic selenium due to sulfur deficiency during the fermentation process. Due to the chemical similarity between sulfur and selenium, selenium can be integrated into the protein structure of yeast selenium cells, essentially replacing the sulfur element. After the inorganic selenium outside the cells is removed by water washing, yeast organic selenium can be produced.

Yeast selenium has been proven to be much safer, more stable, more easily absorbed, more effective and less polluting than inorganic selenium, and it has multiple health functions. Therefore, in recent years, its role in animal nutrition has gradually been recognized, and its application in animal husbandry and fishery production is becoming increasingly widespread. The yeast selenium used in this experiment was provided by Zhengzhou Huafeng Food Technology Co., Ltd., Zhengzhou, China, with a selenium content of 2000 mg/kg.

The experiment studied the effects of yeast selenium on the production performance, nutrient digestibility, gastrointestinal microflora, serum physiological indicators and expression of selenoprotein in the antler of sika deer during the antler growing period, and it explored the appropriate amount of selenium to be added to the diet of sika deer during the antler growing period, so as to provide a theoretical basis for feeding management in the process of large-scale breeding. At present, there are many studies on selenium in cattle and sheep, mainly focusing on the form and source of selenium. There are nearly no reports on the application of yeast selenium at the velvet antler growth stage of sika deer. In this study, the effects of yeast selenium on fiber digestibility, rumen fermentation, digestive enzyme activity and the abundance and function of fiber-degrading microflora were explored, so as to elaborate the mechanism of selenium promoting fiber digestion at the velvet antler growth stage of sika deer and provide further theoretical reference for subsequent scientific research.

## 2. Materials and Methods

### 2.1. Animals and Experimental Design

The subspecies of sika deer which was used in this research is the Jilin sika deer. A total of 20 farmed, healthy sika deer with an initial body weight of (98.08 ± 4.93) kg were selected and randomly divided into 4 groups. The experimental animals were 5-year-old male sika deer for the main purpose of velvet production. Each group was fed in a separate sika deer enclosure. One group named SY1 was the normal control group (N) and the experimental groups were named SY2, SY3 and SY4, respectively.

The provided feed amount was determined according to the dietary requirements of sika deer at the velvet antler growth period. During the experiment, the deer were fed 3 times a day with free water. The trial lasted from 23 April 2020 to 18 August 2020, with a pre-trial period of 7 days and an experimental period of 110 days.

The basal diet for this experiment was provided by a sika deer farm belonging to the Institute of Special Animals and Plants Science of the Chinese Academy of Agricultural Sciences. The basal diet and experimental group had the same nutrient content. For the control group without additional yeast selenium, the selenium content was 0.04 mg/kg (mgSe/kg, DM), which was probably from feed ingredients. The experimental groups were supplemented with yeast selenium, and the selenium contents were 0.3, 1.2 and 4.8 mg/kg, respectively (mg Se/kg, DM).

The ingredients composition and nutrient level of the basal diet are shown in Table 1.

### 2.2. Material and Method

Respectively, at the end of the preliminary experiments (8 d), mid-period (30 d) in the velvet antler growth stage, and at the regeneration velvet antler growth stage (90 d), tests were carried out three times; each period digestion experiment was carried out for three days before the morning feeding records were taken for every deer. We removed impurities from deer feces and dried them in the laboratory for follow-up tests. On the 34th day, the experimental deer were anesthetized before feeding in the morning, and 20 mL rumen fluid was extracted by professional technicians using rumen cannula and then divided into 5 mL sample tubes, which were quickly put into liquid nitrogen. After being taken back to the laboratory, the sample tubes were stored at −80 °C for measurement.

The determination and calculation methods of neutral detergent fiber, hemicellulose, cellulose and crude fiber refer to [8,9]. The determination method of acid detergent fiber refers to [10], using an alkaline sodium hypochlorite–phenol spectrophotometer. The pH values were directly measured by pH meter. The alpha-amylase, protease and cellulase activities were determined by a kit provided by Nanjing Jiancheng. Microbial protein was directly determined by a protein quantitative kit (provided by Nanjing Jiancheng Limited Company, Nanjing, China). Ammonia nitrogen was determined according to T/NAIA 004-2020, and volatile fatty acids were determined by gas chromatography according to T/NAIA 005-2020 (standard method from the Chinese Food Association).

Genomic DNA of the samples was extracted by using a TIANGEN Fecal Genome DNA Extraction Kit (DP328). Agarose gel electrophoresis was used to detect the purity and concentration of DNA. Then, appropriate sample DNA was taken and diluted to 1 ng/μL with sterile water in a centrifuge tube. Taking the diluted genomic DNA as a template, the 16S rRNA V3–V4 region was selected for sequencing. PCR products were detected by electrophoresis with agarose gel of 2% concentration. According to the concentration of PCR products, the same amount of samples were mixed, and the PCR products were detected by 2% agarose gel electrophoresis after full mixing. The target bands were recovered using a gel recovery kit provided by QIAGEN. A Truseq^®^ DNA PCR-Free Sample Preparation Kit library construction kit was used for library construction. The constructed libraries were quantified by Qubit and Q-PCR, and qualified libraries were sequenced by NovaseQ6000. The determination methods and instruments were provided by Beijing Novogene Technology Co., Ltd., Beijing, China.

### 2.3. Sample Collection and Analysis

Species annotation analysis was performed using the Mothur method and the SSU rRNA database of Silva138 (http://www.arb-silva.de/, the last accessed date was 26 March 2023) (with thresholds of 0.8~1) [11,12], obtaining taxonomic information and statistics on the community composition of each sample at the level of family, genus and species. The Corr.test function of the psych package in R was firstly used to calculate the Spearman correlation value of species and environmental factors, and its significance was tested [13]. Then, the pheatmap function of the pheatmap package was used for visualization. Tax4Fun and Faprotax were used to annotate the flora functioning.

### 2.4. Statistical Analysis

The test data were collated by Excel 2019. Then, the one-way ANOVA program of GraphPadPrism 8.0 was used for statistical analysis of the data and histogram drawing. *p* < 0.05 indicated significant difference. *p* < 0.01 indicated extremely significant difference. The statistical results were expressed as “mean ± standard deviation”.

## 3. Results

### 3.1. Digestibility of Fiber in Sika Deer under Different Selenium Levels

The results of fiber digestibility are shown in Table 2. On day 8 of the experiment, the digestibility of acid detergent fiber and crude fiber in sika deer in SY1 was significantly higher than that in SY3 (*p* < 0.05), and the digestibility of hemicellulose in sika deer in SY4 was significantly higher than that in SY2 (*p* < 0.05). The digestibility of acid detergent lignin in sika deer in SY4 was significantly higher than that in SY3. On day 30, the digestibility of neutral detergent fiber and acid detergent fiber in sika deer in SY2 was significantly higher than that in SY1 (*p* < 0.01), the digestibility of neutral detergent fiber in sika deer in SY2 was significantly higher than that in SY3 (*p* < 0.05), and the digestibility of hemicellulose in sika deer in SY2 was significantly higher than that in SY3 (*p* < 0.01). The digestibility of cellulose and crude fiber in sika deer in SY2 was extremely significantly higher than those in SY3 and SY4 (*p* < 0.01), and significantly higher than that in the control group (*p* < 0.05). On day 90 of the experiment, the digestibility of neutral detergent fiber, acid detergent fiber and cellulose in sika deer in SY1 was significantly higher than that in SY2 (*p* < 0.05).

### 3.2. Fermentation Parameters of Rumen Fluid of Sika Deer at Velvet Antler Growth Stage under Different Selenium Levels

Rumen fluid fermentation parameters are shown in Table 3. The rumen fluid pH values of SY2 and SY4 were significantly higher than those of SY1 and SY3 at the velvet antler growth stage (*p* < 0.05). The content of total volatile fatty acids in rumen fluid of SY4 was significantly lower than that of SY1 (*p* < 0.05), and significantly lower than those of SY2 and SY3 (*p* < 0.01). The contents of isobutyric acid and isovaleric acid in rumen fluid of sika deer at the velvet antler growth stage of SY4 were significantly lower than those of SY1 (*p* < 0.01). With the increase of selenium concentration, the content ratio of acetic acid to propionic acid in rumen fluid of sika deer at the velvet antler growing stage increased at first, before decreasing and then increasing again, among which the content ratio of SY3 acetic acid to propionic acid was the lowest (*p* > 0.05), while the content of butyric acid in rumen fluid showed a trend of increasing at first and then decreasing, and the content of butyric acid was the highest in SY3 (*p* > 0.05). There were no significant differences in the contents of ammonia-nitrogen, microbial protein, acetic acid, propionic acid, butyric acid and valerate in rumen fluid of sika deer in the velvet antler growing stage among different experimental groups (*p* > 0.05).

### 3.3. Rumen Digestive Enzyme Activities of Sika Deer at Velvet Antler Growth Stage under Different Selenium Levels

The digestive enzyme activities of rumen fluid are shown in Table 4. The rumen fluid protease activity of SY3 was significantly higher than those of SY1, SY2 and SY4 (*p* < 0.01), and the rumen fluid protease activity of SY2 was significantly lower than those of SY1 and SY4 (*p* < 0.05). Different selenium supplemental levels had no significant effects on the activity of alpha-amylase and cellulase in rumen fluid of sika deer (*p* > 0.05).

### 3.4. Rumen Microflora Relative Abundance of Sika Deer at Velvet Antler Growth Stage under Different Selenium Levels

At the family level, the abundance of rumen microflora of sika deer at the velvet antler growth stage under different selenium levels is shown in Figure 1. Statistics showed that the relative abundance of *Muribaculaceae* in rumen fluid of SY3 was significantly higher than that of SY4 (*p* < 0.05). The relative abundance of *Selenomonadaceae* in rumen fluid of SY4 was significantly higher than those of SY1 and SY2 (*p* < 0.05). The relative abundance of *Acidaminococcaceae* in rumen fluid of SY4 was significantly higher than those of SY1 and SY2 (*p* < 0.01) and significantly higher than that of SY3 (*p* < 0.05). The relative abundance of *Prevotellaceae*, *Lachnospiraceae*, *Christensenellaceae* and *Fibrobacteraceae* in the rumen fluid of SY2 was higher than that in the control group, but there was no significant difference (*p* > 0.05).

### 3.5. Analysis on the Difference of Dominant Strain Species under Different Selenium Levels

As can be seen from Figure 2A, at the genus level, *UCG-005* was the dominant flora of rumen fluid of sika deer at the velvet antler growth stage of SY1 (control group). *Lachnospiraceae* spp. and *Prevotella* spp. were the dominant bacteria in rumen fluid of sika deer at the velvet antler growth stage of SY2 (Se supplemented at 0.3 mg/kg). The dominant florae of rumen fluid of sika deer at the velvet antler growth stage of SY3 (Se supplemented at 1.2 mg/kg) were *Prevotellaceae* spp., *Rikenellaceae* spp. and *Candidatus saccharimonas*. As can be seen from Figure 2B, at the genus level, the dominant florae of rumen fluid of sika deer at the velvet antler growth stage of SY4 are *Lachnospiraceae* spp., *Rikenellaceae* spp. and *Christensenellaceae* spp.

Figure 2C shows that, at the species level, the dominant bacteria groups of rumen fluid of SY1 were *Methanobrevibacter* sp. ABM4, *Oscillospira guilliermondill*, *Romboutsia ilealis*, *Turicibacter* sp. (H121) and *Ruminococcus* sp. (HUN007). *Prevotella ruminicola* and *Fibrobacter succinogenes* were the dominant bacteria groups in rumen fluid of sika deer at the velvet antler growth stage of SY2. The dominant bacteria group in rumen fluid of sika deer at the velvet antler growth stage of SY3 was *Bacteroidales bacterium Bact 22*. According to Figure 2D, the dominant florae of rumen fluid of sika deer at the velvet antler growth stage of SY4 waere *Rikenellaceae* spp. (RC9)*, Christensenellaceaec* spp. and *Lachnospiraceae* spp.

### 3.6. Changes in the Abundance of Rumen Fiber Degradation-Related Bacteria in Sika Deer during Velvet Antler Growth with Different Selenium Levels

The abundance of bacteria associated with fiber degradation is shown in Figure 3. The abundance of *F. succinogenes* in rumen fluid of sika deer at the velvet antler growth stage of SY2 was significantly higher than that of SY1 (*p* < 0.05) and significantly higher than those of SY3 and SY4 (*p* < 0.01). The abundance of *fibrobacterota* in rumen fluid of SY2 was significantly higher than those of SY1, SY3 and SY4 (*p* < 0.01). The abundance of *B. fibrisolvens* in rumen fluid of SY3 was significantly higher than that of SY1 (*p* < 0.01) and that of SY2 (*p* < 0.05) in the velvet antler growth stage of sika deer. Different selenium supplemental levels had no significant effect on the abundance of *R. flavefaciens* and *R. albus* in rumen fluid of sika deer at the velvet antler growth stage (*p* > 0.05).

### 3.7. Analysis of Rumen Differential Bacteria of Sika Deer at Velvet Antler Growth Stage under Different Selenium Levels

As Figure 4 shows, the abundance of Prevotella ruminicola in rumen fluid of SY2 was significantly higher than that of SY1 at the velvet antler growth stage (*p* < 0.05). The abundance of Prevotella ruminicota and Fibrobacter succinogenes in rumen fluid of SY2 was significantly higher than that of SY4 at the velvet antler growth stage (*p* < 0.05). The abundance of Rumen Bacterium R-23 in rumen fluid of SY3 was significantly higher than that of SY2 (*p* < 0.05). The abundance of Selenomonas ruminantium in rumen fluid of SY4 was significantly higher than that of SY2 (*p* < 0.05).

### 3.8. Changes of Rumen Microflora Abundance of Sika Deer during Velvet Antler Growth with Environmental Factors Intervention

As Figure 5 shows, at the genus level, rumen fluid pH was significantly negatively correlated with the abundance of Treponema, Anaerovorax and Fibrobacter flora (*) (*p* < 0.05) and was extremely significantly negatively correlated with the abundance of provotellaceae_UCG.003 flora (**) (*p* < 0.01). The abundance of *SP3.E08,* Butyrivibrio, Quinella and Christensenellaceae R.7 group in rumen fluid of sika deer at the velvet antler growth stage was significantly positively correlated with the supplemental level of yeast selenium (*p* < 0.05). There was an extremely significant positive correlation between the abundance of Succiniclasticum in rumen fluid and the supplemental amount of yeast selenium (*p* < 0.01). Fibrobacter, Prevotellaceae UCG.003 and *Prevotellaceae* spp. in rumen fluid of sika deer at the velvet antler growth stage were significantly negatively correlated with the yeast selenium supplemental level (*p* < 0.01). There was a significant negative correlation between the abundance of *Anaerovorax* and *Bacteroides* in rumen fluid and the supplemental level of yeast selenium (*p* < 0.05).

### 3.9. Screening Differential Genes and Proteins in Rumen Microorganisms

As Figure 6 shown, through differential gene analysis of rumen fluid at velvet antler growth stage of sika deer, four differential genes were found in SY1, six differential genes in SY2 and five differential genes in SY4. The proteins corresponding to different genes in rumen fluid of sika deer at velvet antler growth stage in different treatment groups are listed in Table 5. Among them, SY2 differential proteins are 1, 4-dihydroxy-2-naphthoyl-CoA hydrolase, sedoheptulokinase, prephenate dehydrogenase (NADP+) and glutamine amidotransferase/cyclase, flagellar protein FlbC, fusion protein PurCD.

### 3.10. Prediction of Rumen Microflora Function of Sika Deer at Antler Growth Stage under Different Selenium Levels

According to Figure 7A, the rumen fluid flora of SY2 has transport and catabolism, folding, sorting, degradation, metabolism of terpenoids and polyketides, and glycan biosynthesis and metabolism functions. As can be seen from Figure 7B, the rumen fluid flora of sika deer at the velvet antler growth stage has ligninolysis, xylanolysis, aromatic compound degradation and methanogenesis functions.

## 4. Discussion

The rumen is the main organ for digesting cellulose [14], and selenium can play a role in promoting nutrient digestion and absorption by affecting the structure of rumen microflora and the rumen’s internal environment [15]. The digestibility of neutral detergent fiber, acid detergent fiber, hemicellulose, cellulose and crude fiber can be measured to reflect the strength of rumen digestion of fiber [16,17]. Studies have shown that organic selenium can improve the apparent digestibility of neutral detergent fiber in cows and sheep [1,15]. In this experiment, it was found that the apparent digestibility of neutral detergent fiber, acid detergent fiber, cellulose and crude fiber in the experimental group fed 0.3 mg/kg selenium was significantly higher than that in the control group. The addition of selenium yeast can promote the absorption of neutral detergent fiber, acid detergent fiber, cellulose and crude fiber, which may be because selenium can protect the population of ciliates in the rumen [18], stimulate the secretion of digestive enzymes in the rumen and improve the activity of microorganisms in the rumen [19]. Studies have shown that supplementation with yeast selenium can improve the activity of fiber-degrading bacteria [20] and improve the digestibility of cellulose and hemicellulose in dairy cows [21]. This is because fiber-degrading bacteria in the rumen promote the release of cellulase, which can degrade cellulose [22], unlock the chemical bond between cellulose and hemicellulose [23], expose lignin and facilitate the conversion of crude fibers into microbial proteins in the rumen for the body to use [20].

The sika deer were fed a diet which was fermenting to degrade undigestible nutrients into easily digestible nutrients in the rumen [24]. The optimal pH value of rumen fluid is 5.6~7.5 [25], and the digestibility of cellulose will increase with the increase of the pH value [26,27]. Dietary selenium can promote rumen fermentation [28]. Studies have shown that feeding with organic selenium can effectively improve the rumen fermentation function of dairy cows [20], and ammonia nitrogen, total volatile fatty acids, propionic acid, acetic acid and butyric acid in diets supplemented with yeast selenium have a tendency to increase [29]. It has been reported that feeding with 0.3 mg/kg yeast selenium significantly increased the concentrations of propionic acid and total volatile fatty acids in rumen fluid of beef cattle, dairy cows and goats [2]. This study found that adding selenium content to foods via 0.3 mg/kg of yeast selenium improved rumen acetic acid, propionic acid and the trend of total volatile fatty acid content; with selenium concentrations increased, the rumen liquid pH value increased significantly, and the ammonia nitrogen concentration also showed an increased trend, but the microbial protein, butyric acid, isobutyric acid, valeric acid and isovaleric acid concentrations appeared to decline. This may be because an appropriate selenium concentration can improve the antioxidant function of rumen microorganisms [19], which is conducive to the proliferation of microorganisms [30], and thus improve the yield of total volatile fatty acids. However, a high concentration of selenium inhibits microbial activity and has a toxic effect on rumen microorganisms [31].

Selenium can reduce oxidative stress damage to the rumen and intestinal microorganisms, improve the abundance of microflora related to fiber degradation and the activity of digestive enzymes, promote the secretion of digestive enzymes by rumen microorganisms [32] and increase the degradation rate of cellulose in fiber diets [33]. Studies have shown that selenium can promote rumen Ruminococcus albus, Ruminococcus flavefaciens, Fibrobacter succinogenes, Butyrivibrio fibrisolvens and Fibrobacterota to secrete cellulase and degrade fiber to acetate. It was found that the increase of digestibility of neutral detergent fiber and acid detergent fiber in the body was positively correlated with the concentration of acetate [34]. Selenium can also promote the proliferation of Butyrivibrio fibrisolvens and lactobacillus amylophilus, thereby increasing propionate production [35]. Both the growth and methane production of methanogens require hydrogen ions, but with the addition of selenium, rumen microorganisms produce a large amount of propionate, which can combine with hydrogen ions in rumen fluid, thus inhibiting the activity of methanogens [36]. Selenium can increase the number of rumen protozoa, Butyrivibrio fibrisolvens and Ruminobacter amylophilus [37], thereby promoting the secretion of protease and alpha-amylase [38]. The results showed that dietary yeast selenium containing 0.3 mg/kg selenium could significantly increase the abundance of *fibrobacterota* and *Fibrobacter succinogenes* and their alpha-amylase activity in rumen fluid. The dominant microflora in the rumen would change with different concentrations of selenium, which also explained the inconsistency of microflora abundance under different concentrations of selenium.

In this study, 16S rRNA gene sequencing technology was used to further elaborate the mechanism of selenium promoting rumen fiber digestion in sika deer during the velvet antler growth. The results showed that at the genus level, *Lachnospiraceae NK3A20* group and *Prevotella* were the dominant florae in the rumen fluid of the SY2 group. At the species level, *Prevotella ruminicola* and Fibrobacter succinogenes were the dominant florae in the rumen fluid of sika deer at the velvet antler growth stage of the SY2 group, which was verified by comparative analysis of different flora. The results showed that the yeast selenium supplemental level had a significant positive correlation with *Succiniclasticus*, *Butyrivibrio*, *Christensenellaceae r.7* group, *Quinella* and *sp3.e08*. Due to the close binding of polysaccharides in plant cell walls with cellulose, lignin and pectin, degradation of plant fibers is reduced [39]. Studies have shown that changes in dietary structure can cause changes in the expression of active carbohydrate enzyme genes of *Prevotella* [40]. Rumen *Prevotella ruminicola* can encode similar polysaccharide utilization loci with glycohydrolases [41] and participate in the degradation of hemicellulose through hydrolysis of xylan [42], and the degradation products are mainly succinate, acetate and propionate [43,44]. Studies have shown that *Fibrobacter succinogenes* can promote the degradation of hemicellulose by participating in the encoding of the acetylxylanesterase gene [45]. *Fibrobacter succinogenes* is mainly adsorbed on peptidoglycan by the outer membrane OmpA family protein [46], and hydrolyzed cellulose produces succinate, acetate and formate under the action of cellulase [47,48]. Selenium can mediate the growth process of *Fibrobacter succinogenes* cells by replacing the sulfur atom in L-cysteine, thus affecting the degradation rate of cellulose [49,50].

Through pathway analysis and function prediction, it was found that there were four specific enzymes in the rumen fluid of the SY2 group compared with other groups. These enzymes are 1,4-dihydroxy-2-naphthoyl-CoA hydrolase, sedoheptulokinase, prephenate dehydrogenase and glutamine amidotransferase/cyclase. The analysis of microbial differences showed that the rumen microorganisms of SY2 were significantly different from those of other groups in terms of lignin lysis, xylan decomposition, polysaccharide biosynthesis and metabolism, degradation of aromatic compounds and metabolism of terpenoids and polyketones. These differences could account for the higher fiber digestibility in the SY2 group compared to the other groups. This may be because the intake of yeast selenium leads to changes in *Prevotella ruminicola* and *Fibrobacter succinogenes* in the rumen of sika deer at the velvet antler growth stage. *Prevotella ruminicola* and *Fibrobacter succinogenes* regulate the gene expression level of fiber-degrading enzymes by mediating catabolite repression [46] and promote the high expression of acetylxylanesterase and cellulase. Finally, fiber is degraded into short chain fatty acids. At the same time, the metabolism of terpenoids and polyketones in the SY2 group was enhanced. The short-chain fatty acids in the rumen were condensed to form long-chain fatty acids which could be used by sika deer at the velvet antler growth stage. These metabolic processes accelerated the conversion efficiency of fibrous substances and finally led to higher digestibility of fibrous substances in the SY2 group than in other groups.

## 5. Conclusions

In conclusion, the intake of yeast selenium can increase the abundance of *Prevotella ruminicola* and *Fibrobacter succinogenes* in the rumen of sika deer during the antler growth stage. Furthermore, it can promote the secretion of enzymes such as hydrolase, transferase, oxidoreductase and synthase by mediating the catabolite repression process, thus improving the degradation of fibrous substances.

## Figures and Tables

**Figure 1 microorganisms-11-01444-f001:**
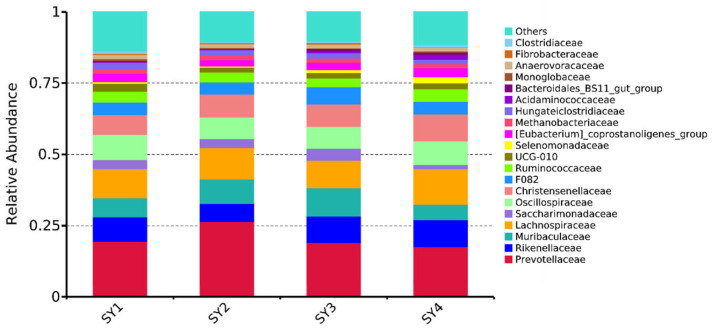
The relative abundance of rumen microorganisms at the family level.

**Figure 2 microorganisms-11-01444-f002:**
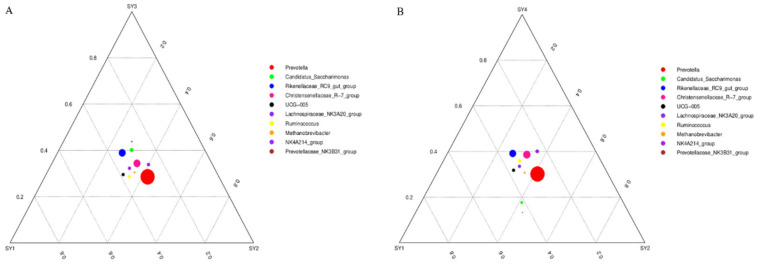
Comparison of dominant bacteria genera and species of sika deer at velvet antler growth stage under different selenium levels. Note: The three vertices in the figure represent the three sample groups, and the circle represents the species. The size of the circle is proportional to the relative abundance. The closer the circle is to a vertex, the higher the content of the object in this group. (**A**,**B**) is the comparison of dominant bacteria at genus level. (**C**,**D**) is the comparison of dominant bacteria at species level.

**Figure 3 microorganisms-11-01444-f003:**
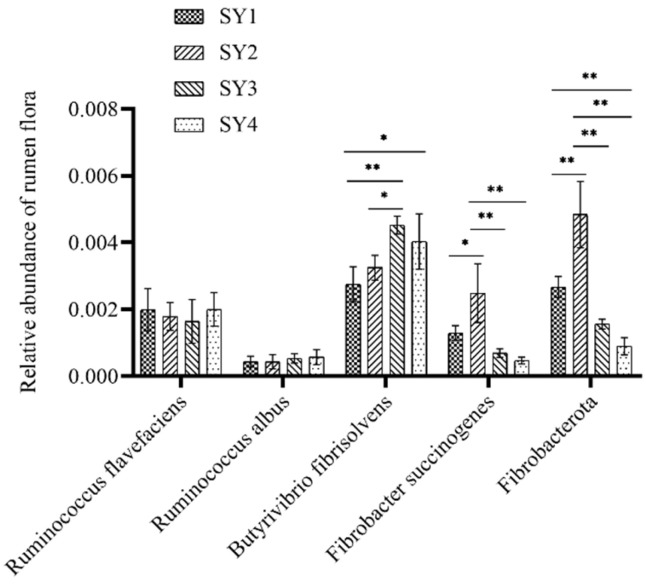
Rumen fluid fibrous degradation related flora. *: *p* < 0.05, **: *p* < 0.01, The same as below.

**Figure 4 microorganisms-11-01444-f004:**
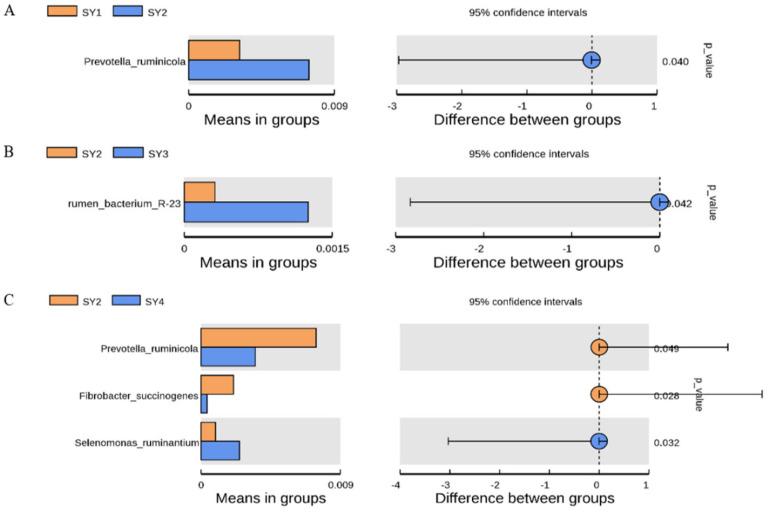
The difference of bacteria between the groups. Note: The figure on the left shows the abundance of species with differences between groups. Each bar in the figure represents the mean value of species with significant differences in abundance between groups in each group. The right graph shows the confidence of inter-group differences. The leftmost endpoint of each circle in the graph represents the lower 95% confidence interval of the mean difference, and the rightmost endpoint of each circle represents the upper 95% confidence interval of the mean difference. The center of the circle is the difference of the mean. The groups represented by the circle color are the groups with the highest mean value. At the far right of the results are the *p* values of the inter-group significance test for the species that differ. (**A**) shows the difference flora analysis of SY1 and SY2. (**B**) shows the difference flora analysis of SY2 and SY3. (**C**) shows the difference flora analysis of SY2 and SY4.

**Figure 5 microorganisms-11-01444-f005:**
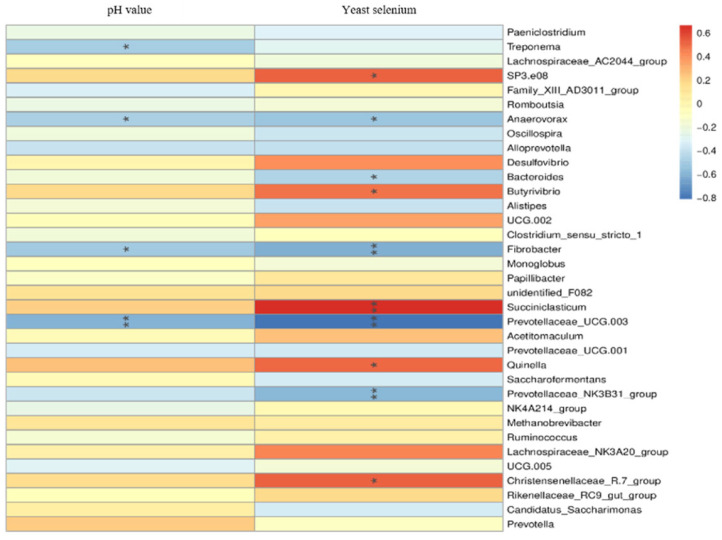
Effects of different environmental factors on the abundance of rumen microflora at genus level.

**Figure 6 microorganisms-11-01444-f006:**
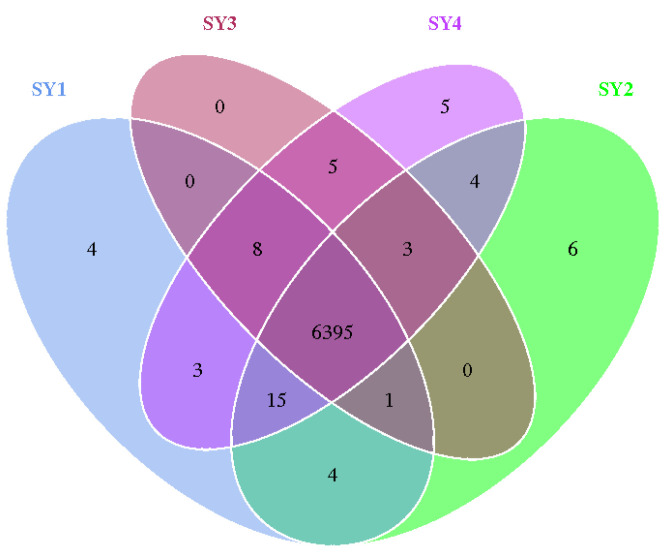
Analysis of rumen microflora differential gene of sika deer at velvet antler growth stage under different selenium levels.

**Figure 7 microorganisms-11-01444-f007:**
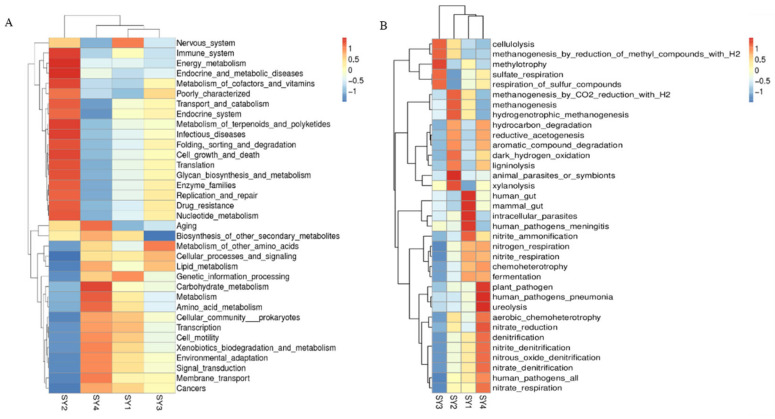
Comparative analysis of rumen microflora function under different selenium levels. Note: (**A**) is the cluster analysis of relative abundance of function using Tax4Fun. (**B**) is the cluster analysis of relative abundance of function using FAPROTAX.

**Table 1 microorganisms-11-01444-t001:** The composition and nutrient levels of the basal diet (air-dry basis) %.

Ingredients	Content
Corn	42.57
Soybean meal	25.54
Maize germ	8.51
Wheat bran	5.96
Silage corn stalks	13.86
CaHPO_4_	1.28
NaHCO_3_	0.43
NaCl	0.85
Premix ^(1)^	1
Total	100.00
Nutrient level ^(2)^	
Dry matter	89.72
Metabolic energy/(MJ/kg)	10.50
Crude protein	19.07
Ether extract	2.31
Neutral detergent fiber	60.41
Acid detergent fiber	16.62
Calcium	0.78
Phosphorus	0.58
Selenium/(mg/kg)	0.04

^(1)^ The premix provided the following per kg of diets: FeSO_4_·H_2_O 0.2 g, ZnSO_4_·H_2_O 0.15 g, MnSO_4_·H2O 0.06 g, CuSO_4_ 0.025 g, VA 4 000 IU, VD 1 000 IU, Thr 0.6 g, Met 1 g, Lys 2 g, Peptides streptozotocin 1 g, Butyrin (50%) 1 g, Natural essential oil 0.5 g, Prebiotics for mildew free 0.7 g, antioxidant 0.1 g. ^(2)^ Metabolic energy is calculated. The calculation formula is below, ME(MJ/kg) = 0.82 × DE, DE(MJ/kg) = 0.209 × CP% + 0.322 × EE% + 0.084 × CF% + 0.002 × NFE%^2^ + 0.046 × NFE% − 0.627. ME was metabolic energy of the diet. DE was digestible energy of the diet. The other nutrient levels were measured values.

**Table 2 microorganisms-11-01444-t002:** Effects of different selenium levels on fiber digestibility of Sika deer %.

Items	Time(d)	Group				*p* Value
		SY1	SY2	SY3	SY4	
Neutral detergent fiber (NDF)	8	73.82 ± 1.75	70.92 ± 0.92	70.52 ± 2.09	73.77 ± 1.61	0.0154
	30	75.66 ± 0.61 ^Bb^	79.42 ± 1.15 ^Aa^	73.98 ± 1.51 ^Bb^	75.84 ± 1.41 ^Bb^	<0.0001
	90	79.92 ± 1.11 ^Aa^	77.04 ± 1.76 ^Ab^	78.46 ± 1.35 ^Aab^	76.79 ± 1.05 ^Ab^	0.0266
Acid detergent fiber (ADF)	8	45.76 ± 2.84 ^Aa^	41.41 ± 2.54 ^Aab^	39.57 ± 3.79 ^Ab^	45.00 ± 2.85 ^Aab^	0.0331
	30	49.09 ± 1.37 ^Bb^	55.78 ± 1.84 ^Aa^	46.70 ± 1.23 ^Bb^	48.20 ± 1.01 ^Bb^	<0.0001
	90	54.47 ± 1.95 ^Aa^	50.29 ± 1.95 ^Ab^	51.79 ± 1.71 ^Aab^	50.24 ± 2.03 ^Aab^	0.0300
Hemicellulose	8	84.47 ± 1.36 ^Aab^	82.11 ± 0.37 ^Ab^	82.26 ± 1.48 ^Aab^	84.69 ± 1.27 ^Aa^	0.0099
	30	85.75 ± 0.53 ^ABab^	88.39 ± 1.11 ^Aa^	84.34 ± 1.89 ^Bb^	86.33 ± 1.82 ^ABab^	0.0094
	90	89.58 ± 1.20	87.18 ± 1.78	88.59 ± 1.24	86.87 ± 0.76	0.0519
Cellulose	8	55.53 ± 3.30	49.28 ± 2.77	49.66 ± 6.15	51.55 ± 2.94	0.1523
	30	59.34 ± 1.03 ^Bb^	66.49 ± 1.40 ^Aa^	54.86 ± 0.92 ^Cc^	55.61 ± 1.75 ^Cc^	<0.0001
	90	63.13 ± 2.86 ^Aa^	57.93 ± 2.56 ^Ab^	59.81 ± 1.57 ^Aab^	56.91 ± 2.22 ^Ab^	0.0172
Crude fiber (CF)	8	51.99 ± 5.30 ^Aa^	49.04 ± 4.62 ^Aab^	37.14 ± 7.19 ^Ab^	43.88 ± 3.02 ^Aab^	0.0270
	30	51.65 ± 4.95 ^ABb^	63.25 ± 5.86 ^Aa^	42.56 ± 4.99 ^Bb^	45.99 ± 4.14 ^Bb^	0.0008
	90	53.30 ± 5.07	53.43 ± 1.32	54.36 ± 4.03	54.24 ± 8.52	0.6724
Acid detergent lignin	8	41.70 ± 6.24 ^ABab^	41.01 ± 4.41 ^ABab^	35.85 ± 6.36 ^Bb^	51.24 ± 4.81 ^Aa^	0.0133
	30	45.53 ± 4.94	54.87 ± 5.98	48.61 ± 7.43	53.61 ± 1.49	0.0895
	90	59.60 ± 3.48	56.27 ± 4.58	57.36 ± 3.93	59.40 ± 3.77	0.6109

In the same row, values with no letter or the same letter superscripts mean no significant difference (*p* > 0.05), while with different small letter superscripts mean significant difference (*p* < 0.05), and with different capital letter superscripts mean significant difference (*p* < 0.01). The same as below.

**Table 3 microorganisms-11-01444-t003:** Effects of different selenium levels on rumen fermentation parameters of sika deer.

Items	Groups				*p* Value
	SY1	SY2	SY3	SY4	
pH value	7.21 ± 0.09 ^Ab^	7.32 ± 0.09 ^Aab^	7.21 ± 0.14 ^Ab^	7.52 ± 0.13 ^Aa^	0.0177
NH_3_-N/(mg·dL^−1^)	47.83 ± 6.17	44.88 ± 5.67	43.33 ± 3.68	48.75 ± 6.40	0.6977
MCP/(mg·mL^−1^)	2.80 ± 0.75	1.86 ± 0.47	1.98 ± 0.06	1.81 ± 0.44	0.0746
TVFA/(mmol·L^−1^)	18.62 ± 2.96 ^ABab^	23.59 ± 2.78 ^Aa^	22.67 ± 1.84 ^Aa^	13.17 ± 1.15 ^Bb^	0.0009
Acetate/%	11.94 ± 1.86 ^ABbc^	16.07 ± 2.00 ^Aa^	15.09 ± 1.09 ^Aab^	8.70 ± 0.72 ^Bc^	0.0006
Propionate/%	3.02 ± 0.46 ^ABbc^	4.05 ± 0.50 ^Aa^	3.99 ± 0.42 ^Aab^	2.13 ± 0.13 ^Bc^	0.0004
Acetate/Propionate	3.95 ± 0.14	3.97 ± 0.24	3.80 ± 0.13	4.09 ± 0.18	0.4126
Isobutyrate/%	0.58 ± 0.08 ^Aa^	0.53 ± 0.05 ^Aab^	0.55 ± 0.01 ^Aa^	0.40 ± 0.05 ^Ab^	0.0124
Butyrate/%	2.07 ± 0.46 ^Aab^	2.11 ± 0.25 ^Aa^	2.13 ± 0.32 ^Aab^	1.29 ± 0.24 ^Ab^	0.0294
Isovalerate/%	0.76 ± 0.11 ^Aa^	0.59 ± 0.07 ^ABab^	0.65 ± 0.01 ^ABab^	0.48 ± 0.07 ^Bb^	0.0093
Valerate/%	0.26 ± 0.04	0.24 ± 0.04	0.26 ± 0.02	0.18 ± 0.03	0.0588

In the same row, values with no letter or the same letter superscripts mean no significant difference (*p* > 0.05), while with different small letter superscripts mean significant difference (*p* < 0.05), and with different capital letter superscripts mean significant difference (*p* < 0.01). The same as below.

**Table 4 microorganisms-11-01444-t004:** Effects of different selenium levels on digestive enzyme activity in rumen fluid of sika deer.

Items	Groups				*p* Value
	SY1	SY2	SY3	SY4	
α-amylase/(U·dL^−1^)	50.84 ± 1.40	52.15 ± 2.15	49.81 ± 2.28	51.55 ± 1.31	0.4784
Protease/(U·mL^−1^)	3.15 ± 0.30 ^Bb^	2.38 ± 0.11 ^Bc^	3.97 ± 0.22 ^Aa^	3.08 ± 0.17 ^Bb^	0.0003
Cellulase/(U·mg^−1^)	34.71 ± 1.29	24.15 ± 1.30	30.01 ± 6.61	26.34 ± 8.03	0.2793

In the same row, values with no letter or the same letter superscripts mean no significant difference (*p* > 0.05), while with different small letter superscripts mean significant difference (*p* < 0.05), and with different capital letter superscripts mean significant difference (*p* < 0.01). The same as below.

**Table 5 microorganisms-11-01444-t005:** Analysis of rumen flora differential protein of sika deer at velvet antler growth stage under different selenium levels.

Groups	KO Number	Enzyme/Protein	Enzyme Classification
SY1	K11389	glyceraldehyde-3-phosphate dehydrogenase (ferredoxin)	EC:1.2.7.6
	K02092	allophycocyanin alpha subunit	
	K02975	small subunit ribosomal protein S25e	
	K07547	2-[hydroxy(phenyl)methyl]-succinyl-CoA dehydrogenase BbsC subunit	EC:1.1.1.35
SY2	K12073	1,4-dihydroxy-2-naphthoyl-CoA hydrolase	EC:3.1.2.28
	K11214	sedoheptulokinase	EC:2.7.1.14
	K00211	prephenate dehydrogenase (NADP+)	EC:1.3.1.13
	K01663	glutamine amidotransferase/cyclase	EC:2.4.2.- 4.1.3.-
	K02384	flagellar protein FlbC	
	K13713	fusion protein PurCD	EC:6.3.2.6 6.3.4.13
SY4	K14331	fatty aldehyde decarbonylase	EC:4.1.99.5
	K12535	outer membrane protein RsaF	
	K12533	ATP-binding cassette, subfamily C, bacterial RsaD	
	K05348	2-hydroxycinnamic acid beta-D-glucosylisomelase	EC:5.2.1.-
	K13585	holdfast attachment protein HfaA	

## Data Availability

Data is unavailable due to privacy or ethical restrictions.

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
