# Peer review of "Effects of Different Yeast Selenium Levels on Rumen Fermentation Parameters, Digestive Enzyme Activity and Gastrointestinal Microflora of Sika Deer during Antler Growth"

_microorganisms, 2023, doi:10.3390/microorganisms11061444_

Round 1

Reviewer 1 Report

The authors present the study of yeast selenium supplementation effects in sika deer on several parameters during antler growth.

In general manuscript is not acceptable in current form - needs more detailed materials and methods section and introduction where it actually describes effects of selenium and takes care of proper referencing.

Language has to be significantly improved. In current form reading the manuscript is difficult and leaves the reader with quite a lot of guessing to do.

Shorten abstract a bit – no need to go into details in abstract about groups. Also remove 0 from line 23 – this should be a control group, right?

Name of species is written in lowercase – e.g. Fibrobacter succinogenes

First paragraph of introduction lacks any references, and partially is repeated in second paragraph.

What Wu 2012 reference refers to? Is this a research article dissertation?

Authors claim selenium supplementation can promote rumen microflora colonization and cite Feirreira et al., 2021 which claims this for humans. It this applicable and transferrable to ruminants?

Reference Faixová et al., 2007 is mentioned in the text and absent from references.

Can't find any mention of reference Fu et al., 2020 online – pubmed and google do not return this article.

Authors fail to mention subspecies of sika deer on which experiment was carried on and do not mention scientific name…

Ln 87-88 – does this also mean experimental groups had 0.04 mg/kg of Se in basal diet – unclear from the text. In what form was selenium in basal diet?

Method section is poorly written – only clues were given and combined with hard to follow language leaves reader unable to fully understand what and how was done. E.g. authors reference few methods as T/NAIA. What is that? Couldn’t find any reference that was in English.

What was the yield of sequencing for each sample, what quality controls were done, more detail is needed.. Why sequences were not deposited to a public database?

There is mention of figures in the text and they are not part of the manuscript

Results and conclusions seem ok but without proper materials and methods section are unconvincing.

Author Response

More detailed materials and methods in  introduction section has been added and the effects of selenium was introduced.

Reviewer 2 Report

The article is well structured and with ample indications both in the introduction and in materials and methods and discussion. However. in general I have not been able to understand well the essential purpose of the research carried out.
Furthermore, I have noticed a series of typographical errors which I have reported in the pdf version which I have corrected and which I am attaching.
Finally, in the introduction the authors did not clearly indicate what is meant by the term "yeast selenium". It should be included in the article.
Furthermore, in line 20, the authors report that the deer they used in the tests weighed more than 98 kg. It seems to me a lot, given that the indications provided by the zoological treatises that I have consulted mention, for the sika deer, a weight of 4-44 kg. Furthermore, the authors did not clearly specify that the deer they used in the experimental tests were farmed, even if we can guess it from the article. However, the authors should specify it.

Author Response

The experiment studied the effects of yeast selenium on the production performance, nutrient digestibility, gastrointestinal microflora, serum physiological indicators and the expression of selenoprotein in antler antler of sika deer during antler growing period, and explored the appropriate amount of selenium added in the diet of sika deer during antler growing period, so as to provide a theoretical basis for feeding management in the process of large-scale breeding.

Reviewer 3 Report

Your manuscript deals with differences in rumen fermentation parameters, digestive enzyme activity and the gastrointestinal microflora of sika deer after supplementation with different levels of yeast selenium. Your study shows interesting changes especially for researcher focusing on sika deer and people working on micro biotic changes due to nutrition (in animals). Despite the mentioned observations, your manuscript needs a further revision. Attached you will find some major and minor issues regarding your project.

Major Issues:

1.     A linguistic proof reading by a professional for English language/native speaker is recommended.

2.     Please create a separate material & method section.

3.     Please describe or cite the SDS-based method for DNA extraction.

4.     Very confusing Table 2. Please improve.

5.     Please specify the groups mentioned in section 3.5.

6.     Please introduce “Yeast selenium”.

7.     Please mention potential impacts on the antler growth if evaluated. Please further explain why the part of your title “during antler growth” is important for your study.

Minor Issues:

8.     A figure regarding 16rRNA gene sequencing would be beneficial.

Author Response

The manuscript will be checked by professional for English language/native speaker 

Round 2

Reviewer 1 Report

The authors have made substantial improvements to the manuscript. Still some issues remain. There are still no figures for this manuscript. None are in the manuscript itself and only tables are in supplementary file. Naming of figures in manuscript is inconsistent - they are named using letters and numbers - e.g. Fig. C and Fig. 3 are used.

Bacterial species are still written with capital letter in lines 400-454 e.g. Prevotella Ruminicola.

Author Response

Response 1there are total 7 figures in this manuscript named fig1~fig7 as follows, and the figures have oploaded and emailed to the editor.Thanks a lot for your advice. There is a mistake in supplementary file.
Naming of the figures in manuscripts are  from Fig.1  to Fig.7. Fig 2 contains 4 figures names Fig 2A、Fig 2B、Fig 2C and Fig 2D.
Fig. 4 contains 3 figs named fig.4A、fig.4B and fig.4C.

Reviewer 2 Report

I apologize, I thought I had already responded to your request for a second evaluation of the revised form of the article. I have read the new version and I believe that the text is now clearer and more complete and I have NO other comments to make. Congratulations to the authors.

Author Response

Thank you a lot for my manuscript